# Interaction between Gut Microbiota and Dendritic Cells in Colorectal Cancer

**DOI:** 10.3390/biomedicines11123196

**Published:** 2023-12-01

**Authors:** Kawther Zaher, Fatemah Basingab

**Affiliations:** 1Immunology Unit, King Fahad Medical Research Centre, King Abdulaziz University, Jeddah 21859, Saudi Arabia; 2Department of Medical Laboratory Sciences, Faculty of Applied Medical Sciences, King Abdulaziz University, Jeddah 21859, Saudi Arabia; 3Department of Biological Sciences, Faculty of Science, King Abdulaziz University, Jeddah 21585, Saudi Arabia

**Keywords:** C-type lectin receptor, CRC, DCs, interleukins, microbiota, metabolic syndrome

## Abstract

Colorectal cancer (CRC) is a malignancy that manifests in serial stages and has been observed to have an escalating incidence in modern societies, causing a significant global health problem. The development of CRC is influenced by various exogenous factors, including lifestyle, diet, nutrition, environment, and microbiota, that can affect host cells, including immune cells. Various immune dysfunctions have been recognized in patients with CRC at different stages of this disease. The signature of microbiota in the development of CRC—inflammation related to obesity, diet, and reactive host cells, such as dendritic cells (DCs)—has been highlighted by many studies. This study focuses on DCs, the primary cellular mediators linking innate and adaptive immune responses against cancer. In addition, this review focuses on the role of microbiota in dysbiosis and how it affects DCs and, in turn, the immune response and progression of CRC by stimulating different sets of T cells. Additionally, DCs’ role in protecting this delicate balance is examined. This is to determine how gene yields of commensal microbiota may be critical in restoring this balance when disrupted. The stages of the disease and major checkpoints are discussed, as well as the role of the C-type lectin receptor of immature DCs pattern recognition receptor in CRC. Finally, based on a thorough examination of worldwide clinical studies and recent advancements in cancer immunotherapy, it is recommended that innovative approaches that integrate DC vaccination strategies with checkpoint inhibitors be considered. This approach holds great promise for improving CRC management.

## 1. Introduction

Colorectal cancer (CRC) significantly contributes to cancer-related deaths and ranks as the third most prevalent cancer globally [1]. CRC is notably high among populations that adopt a Western lifestyle; it is also rising in other locations, primarily low-income nations, creating a worldwide health threat [2]. CRC develops slowly over years, as adenomas originate from tiny, hard-to-see neoplastic foci into malignant carcinomas that may spread throughout the body [3]. CRC is reported to have considerable heterogeneity due to genetic instability [4,5]. It may also be caused by lifestyle, food, nutritional intake, environmental circumstances, and the microbiome. These variables affect non-neoplastic cells, including immune cells, increasing heterogeneity [6,7]. CRC is caused mainly by immune system malfunction, as it begins by impairing the host’s anti-tumor immunity, the so-called tumor microenvironment [8]. The tumor microenvironment changes early in neoplastic transformation and progression to promote cell proliferation. Tumor development and metastasis or immune-mediated cancer inhibition may follow. Inflammatory and immunological cells in the tumor microenvironment may accelerate colorectal cancer. These cells may limit tumor development or cause chronic inflammation, suppressing the immune system and promoting CRC progression [9,10]. The purpose of this article is to spotlight the leading causes of CRC and how they affect gut microflora. In addition, this review also focuses on how DCs play an important role in tumor regulation and, eventually, explains the promising role of immunotherapy in CRC treatment and control.

## 2. Causes and Symptoms of CRC

Diet and lifestyle have been associated with CRC risk for decades. Modifiable risk factors cause 50–60% of CRC cases [11]. Tobacco use, excessive alcohol use, obesity, lack of exercise, consuming red and processed meat, and insufficient dietary fiber and other nutrients worsen health (Figure 1). The microbiome, such as bacteria, viruses, and fungi, is essential to health. Microbiota disturbances may cause CRC. According to Song and Chan [12], environmental risk factors may cause and promote CRC by altering the gut microbiota. Personal or family history of colorectal polyps or CRC, Lynch syndrome, inflammatory bowel disease (IBD), racial/ethnic origin, and type 2 diabetes are all unchangeable risk factors for CRC [13]. CRC can develop in persons as young as in their 20s, if they are predisposed to the disease. Advanced age is the most prominent risk factor [14]. Over 90% of new cases and 94% of CRC-related fatalities occur in people over 50. Medium-risk people over 50 should be examined for CRC, and high-risk people should be monitored. High-risk factors include CRC syndromes, adenomatous polyps, IBD, and personal or family history of CRC [15]. Constipation, diarrhea, stomach pain, and rectal pain precede 70–95% of early-onset CRC. Later, anemia, rectal bleeding, and weight loss occur [16].

### 2.1. Effect of Red Meat and Junk Food

Research by the World Cancer Committee [17] demonstrated that the regular consumption of 100 g of red and processed meat heightens the likelihood of developing CRC by 12%. This heightened susceptibility is attributable to the presence of heme iron, sulfur, and choline, in addition to nitrates, nitrites, emulsifiers, and heterocyclic amines and polycyclic aromatic hydrocarbons, which emanate from processing and cooking techniques and may stimulate the onset of cancer [18]. These meals may fuel gut microbes that generate carcinogens. Red meat provides choline and carnitine. Red meat elevated blood and urine trimethylamine-N-oxide (TMAO) in a random controlled experiment. TMAO increases heart disease and mortality. Two bacterial genes, cutC (choline TMA-lyase) and cutD (choline TMA-lyase-activating enzyme), were discovered in extremely high numbers, supporting the concept that the choline-TMAO pathway may modify metagenomes and cause CRC. *Hungatella hathaway*, *Clostridium asparagiforme*, *Klebsiella oxytoca*, and *E. coli* increased in CRC patients with sequence changes in these two genes. Several prospective studies have linked high dietary and blood choline levels to CRC [19]. DNA methylation and synthesis require choline, folate, and vitamin B12. TMAO only increased CRC risk in vitamin B12-deficient people. Its effects may vary, depending on whether it is in the carbon metabolism cycle or in the stomach for microbial TMAO synthesis [20]. CRC is connected to bacterial secondary bile acids, processed foods, red meat, and free radicals [21]. Bile acid content and concentration variations in metabolic diseases and infectious bowel disease (IBD) enhance CRC risk. Several analyses of eight geographically and methodologically diverse fecal shotgun genomic studies found that higher secondary bile acid was produced [22]. According to a study, high-fat diets may contribute to the epidemic of CRC in young people. It has been found that high-fat diets alter the gut bacteria and alter the digestive substances, called bile acids, in mice. Consequently, inflammation occurs, increasing the chance of developing colorectal cancer, a disease that is notoriously difficult to treat [23]. When the body consumes fat, bile acids are synthesized in the liver, and they play a role in facilitating lipid absorption by the small intestine. Bile acids are reabsorbed and subjected to enterohepatic circulation in the small intestine. Bile acids undergo complex biotransformation in the colon by gut bacteria, resulting in secondary bile acids that promote tumor growth. Excessive dietary fat intake leads to a high level of secondary bile acids in feces and primes the gut microbiota to produce bile acids. It is believed that this promotes an altered overall bile acid pool, resulting in an altered intestinal and hepatic cross-signaling of the farnesoid X receptor (FXR), the receptor for bile acids. The FXR gene is a main regulator of bile acid-mediated effects on intestinal tumorigenesis, integrating dietary, microbial, and genetic risk factors. In healthy and tumorigenic conditions in the intestine, selective FXR agonist or antagonist activity is determined by additional factors such as bile acid concentration, composition, and genetic instability of the cells [24].

### 2.2. Correlation between Healthy Diet and CRC

Many epidemiological inquiries remain to be conducted to substantiate this concept. Based on a meta-analysis of 21 studies, there appears to be no link between fiber consumption and CRC risk. Nevertheless, the European Prospective Investigation into Cancer and Nutrition (EPIC) cohort has persistently demonstrated that a higher fiber intake can lower susceptibility to CRC [25]. Europeans eat cereals, but Americans eat fruits and vegetables, which may explain the observed behavior. US cohorts stated that low fiber consumption might have prevented efficacy. According to numerous studies, whole grains reduce the risk of CRC. A meta-analysis found that increasing a whole grain diet by 90 g/day reduced CRC risk by 17% [17].

### 2.3. Role of Obesity and Physical Activity on CRC

A meta-analysis revealed that a rise in body mass index (BMI) of 5 kg/m^2^ raised the risk of CRC by 5%. Additionally, obesity may potentially escalate the incidence of early-onset CRC [26]. Obesity is linked to inflammatory and metabolic changes in adipokines, insulin-like growth factor 1 signaling, systemic inflammation, sex hormones, and CRC risk. Obesity-induced gut microbiota alterations and metabolic byproducts may affect cancer development. Obesity decreases gut microbiota diversity [27]. 

Obesity-induced intestinal barrier dysfunction may worsen systemic inflammation. Leaking microbiota products such as lipopolysaccharide (LPS) causes metabolic endotoxemia. Greater BMIs are related to greater LPS-binding protein (LBP) and LPS levels. Losing weight decreases blood LBP and LPS [28]. Adenomas were associated with increased LPS levels in cross-sectional research. Villous adenomas had higher LPS levels than tubular ones. An LBP gene polymorphism (rs2232596) has been associated with CRC [29].

According to cohort data, high exercisers decrease CRC risk by 19% compared with non-exercisers. Exercise has not been linked to rectal cancer. In many cross-sectional studies, exercise influences the gut microbiome’s composition and function [17].

A 6-week intervention of endurance-based exercise was provided to a population of 32 people who lived sedentary lifestyles, after 2 weeks of the baseline study. The subjects then had a 6-week exercise-free washout period. The study found that only skinny participants had short-chain fatty acid concentrations in their stools after exercise. The number of butyrate-producing bacterial taxa increased, linked to body composition changes in lean people [30].

### 2.4. The Role of GUT Microbiota in Dysbiosis and CRC 

Specific nutritional components can regulate the gut microbiome and promote a persistent inflammatory state by modulating immune and inflammatory pathways, making diet a significant factor in CRC onset, progression, and prevention. Human microbiota includes bacteria in the gastrointestinal, genitourinary, oral, respiratory, and cutaneous systems, among others. These bacteria interact with the body in various mechanisms and are essential for several physiological routes, including immunology, tissue growth, nutrition absorption, metabolism, and cancer [31]. Recent research showed that nutritional, genetic, and environmental factors affect the microbiome. Gut microorganisms also affect food metabolism. Nutrition affects microbiota in populations with various diets. According to the study, “healthy” diets like the Mediterranean diet increase microbial biodiversity. Dietary fiber, polyphenols, and lipids are crucial food–gut microbiome interactions. Different dietary lipids may affect gut microbiota variety, microorganisms, and activity. Metabolic effects may modulate systemic low-grade inflammation. Many pathways link dysbiosis to illnesses, including cancer [32]. Oncogenesis may also result from microbiota-induced mucosal inflammation or systemic metabolic and immunological disruption. The microbiome may indirectly alter cancer treatment and immunity. The inhibition of programmed cell death (PD-1) is associated with antitumor effects of epithelial malignancies in hepatocellular carcinoma and is also linked to *Akkermansia muciniphila* frequency [33]. Lower levels of this bacterium in rats and humans are connected to obesity, insulin resistance, type 2 diabetes, and other cardiometabolic diseases. CRC patients’ fecal and mucosal microbiota change during the disease, reducing bacterial diversity [34].

It is important to note that LPS is an important component of the outer layer of bacteria and has a strong pathogenic effect [35]. Inflammation and immune responses in the hosts can be triggered through TLR activation (one of the pattern recognition receptors), among which TLR4 and TLR2 are the most important receptors [36]. The FimA and Mfa1 subunits of bacterial fimbriae are important in promoting bacterial adhesion to host tissues and their invasion [37]. A TLR on endothelial cells [38], macrophages [39], and DCs [40] can also recognize it, activating the cells and causing them to produce cytokines and adhesion molecules. These cytokines stimulate the proliferation and differentiation of DCs. A link between innate and adaptive immunity is established when inflamed endothelial cells trigger macrophages and immature tissue DCs to present the bacterial cell wall on their surface for T cells called major histocompatibility molecules (MHC-I). DCs recognize pathogen-associated molecular patterns (PAMP) and damage-associated molecular patterns (DAMP) [41]. As DCs mature, metabolic, cellular, and gene transcription programs are activated, allowing them to migrate from peripheral tissues to secondary lymphoid organs, where T lymphocyte-activating antigen presentation may occur in response to CCL19 and CCL21 [42]. The maturation of DCs is characterized by the loss of adhesive structures, the reorganization of the cytoskeleton, and an increase in motility. As a result of DC maturation, the level of endocytic activity declines; however, the expression of MHC-II and co-stimulatory molecules such as CD80 and CD86, as well as the chemokine receptor CCR7, is increased, as are the levels of pro-inflammatory cytokines such as TNF-α and IL-12 [43]. A mature DC expresses higher levels of the chemokine receptor, CCR7 [44], and secretes cytokines required for T-cell activation [45]. Consequently, mature DCs interact with antigen-specific T cells to trigger antigen-specific immune responses [46]. By interacting with CD4^+^ T cells (called MHC-II), DCs may induce differentiation into different T helper (Th) cells [47], Th2 cells [48], Th17 cells [49], or other CD4^+^ T cells [50]. This way, DCs can trigger responses against intracellular antigens from various cell types [51] and even prime CD8^+^ lymphocytes without CD4^+^ T cells [52]. Cross-presentation also introduces tolerance to intracellular self-antigens not expressed by APC, called cross-tolerance [53]. The “immature” cells are, however, extremely effective at capturing antigens due to their high endocytic capacity via receptor-mediated endocytosis, including lectins [54], Toll-like receptors [55], FC receptors, and complement receptors [56]. Thus, immature DCs act as sentinels against invading pathogens and as tissue scavengers, capturing apoptotic and necrotic cells [57]. Accordingly, immature DCs are essential for inducing and maintaining immune tolerance [58]. DCs internalize cell apoptosis and the loss of critical details in polyps, such as tissue turnover. However, they do not induce DCs maturation [59]. In this way, their antigens are presented to T cells without the activating co-stimulatory signals delivered by a mature DC, resulting in the apoptosis of T cells and the development of regulatory T cells. It has been demonstrated that a “tolerogenic DC” expresses fewer co-stimulatory molecules and proinflammatory cytokines. Still, it upregulates the expression of inhibitory molecules (such as PD-L1 and CTLA-4) and secretes anti-inflammatory cytokines (IL-10, tumor growth factor beta, for example). In this situation, the DC fails to increase the co-stimulatory molecules required to activate T cells, meaning that the immune system leaves uncontrolled disease despite recognizing pathogens or changes in the tumor microenvironment [52,58].

#### 2.4.1. *Fusobacterium nucleatum*

Cohort studies showed that colorectal neoplasia patients have more *Fusobacterium nucleatum* (*F. nucleatum*) in their feces than controls. This was connected to more advanced disease, less T-cell infiltration, a higher risk of recurrence, and a worse prognosis. This was linked to serrated pathway molecular characteristics. The inhibition of Myeloid-derived suppressor cells and natural killer (NK) and T cells enhances tumor growth. Modifying E-cadherin/-catenin may affect this mechanism [35]. According to recent studies, *F. nucleatum* stimulates the production of reactive oxygen species (ROS) and inflammatory cytokines (e.g., IL-6 and TNF) in colorectal cancer [38,60]. MLH1 is epigenetically silenced due to inflammation and ROS, reducing mismatch repair protein’s enzymatic activity (MMR) activity. In addition, virulence factors derived from *F. nucleatum* also inhibit T cell proliferation [33,61]. This may be consistent with a recent study finding that a greater abundance of *F. nucleatum* in colorectal carcinoma tissue was associated with a lower density of T cells in the tumor microenvironment. Based on these findings, *F. nucleatum* may downregulate antitumor T-cell-mediated adaptive immunity [62].

#### 2.4.2. Enterotoxigenic *Bacteroides fragilis* (ETBF)

Among the symbiotic bacteria in the intestinal tract is *Bacteroides fragilis*. The majority of bacteria in the human body aid in the digestion of food and maintain intestinal health. Occasionally, however, these bacteria produce a toxin that disrupts the cells on the surface of the gut, resulting in the development of CRC. B fragilis has two subtypes, nontoxic B fragilis and enterotoxigenic *B. fragilis* (ETBF), which produce *B. fragilis* toxin (BFT). ETBFs produce a 20 kDa metalloprotease, *B. fragilis* toxin (BFT), which disturbs the intestinal environment and promotes inflammation [63]. BFT destroys the epithelial barrier and E-cadherin cleavage. Additionally, cleavage of E-cadherin can activate the Wnt signal transduction pathway, cause mucosal inflammation, and promote colon cancer development. Further, the STAT3 pathway is required to develop Th17 cells and colon transformation. Th17 cells are stimulated to produce IL-17 after ETBF stimulates the STAT3 pathway, which activates the NF-κB and Wnt pathways, creating intestinal inflammatory tumor microenvironments. Through its rapid activation of the STAT3 pathway, ETBF is involved in the production of IL-17 by Th17 cells. Injection of an anti-IL-17 antibody in mice can inhibit tumor formation by inhibiting IL-17 production [64]. ETBF can also upregulate the expression of spermine oxidase (SMO) in colonic epithelial cells, thus increasing the generation of SMO-dependent reactive oxygen species (ROS), causing DNA damage and, ultimately, leading to CRC development [65].

#### 2.4.3. *Escherichia coli* (*E. coli*)

IBD and CRC patients have higher mucosa-associated *E. coli* levels. In CRC, *E. coli* invades the intestinal wall and becomes intracellular. Polyketide synthase gene complex (pks) bacteria generate genotoxin colibactin, more commonly in CRC patients. Later-stage tumors have more *pks+ E. coli* strains. Ki-67 expression correlates with internalized and mucosa-associated *E. coli*. One fecal microbiome investigation found *E. coli* enrichment in CRC patients. *E. coli* prefers to colonize the mucosal lining and, intracellularly, occupy intestinal cells rather than the lumen, impeding feces removal [66].

## 3. Pathogenesis and Genetic Alteration of CRC

CRC originates in pre-cancerous polyps. The term “polyp” refers to localized outgrowths or clusters of aberrant cells that extend into the intestinal cavity from the gut’s mucosal layer. CRC exhibits either sessile or pedunculated polyps [67]. If these polyps have enough genetic alterations, their multiplying cells may be able to permeate the intestinal wall, a characteristic of colorectal cancer. These cells may then alter, move to nearby lymph nodes, and reach distant metastatic locations. When the polyp grows, it may acquire genetic mutations and epigenetic alterations that cause cytologic and histologic dysplasia [68]. Slow cellular DNA damage may cause high-grade dysplasia, which increases the risk of invasive cancer [69]. Without removal, these polyps can invade the colon and rectal wall and spread. New blood arteries allow cancer cells to access the lymphatic and circulatory systems and metastasize to distant organs. Pre-cancerous polyps must be removed quickly to break the adenoma-carcinoma cycle and prevent colorectal cancer. Histologically, genetic and epigenetic alterations cause a polyp to become a malignancy (Figure 2). DNA alteration can be either inherited or acquired. Inherited mutations, including APC (adenomatous polyposis coli), PMS2, MSH2, MLH1and gene changes, cause 5% of CRC cases. Hereditary mutations with spontaneous APC and DNA divergence repair metamorphoses have illuminated the genetic path from premalignant polyps to cancer [70]. Sessile serrated polyps (SSPs) and adenomas polyps often cause CRC. The chromosomal instability route affects 65–70% of sporadic malignancies, which are often linked to conventional adenomas. Many mutations distinguish this method. Two genes cause CRC APC gene mutations to alter chromosomal segregation during cell division, while *KRAS* oncogene changes affect cellular proliferation, differentiation, motility, and survival. Mutations may affect transcription and apoptotic regulator *p53*. This may affect cellular processes and cause cancer [71]. However, *BRAF* gene mutations that disrupt growth signals and impede apoptosis typically cause sessile serrated polyps (SSPs). SSPs have *KRAS* mutations at 23, 21, 24, and 13, but less than adenomatous polyps (Figure 2). Serrated lesions cause gene promoter hypermethylation in CRC [72]. Methylation of the promoter region may inhibit gene transcription and function. Inactivating this gene affects many genes, including development-promoting genes [73]. CpG island methylators contain abnormally methylated genes such as *BMP3* and *NDRG4* [74]. Microsatellite instability (MSI) disrupts DNA repair genes, promoting genetic diversity in CRC. MSI may cause short, non-coding, repetitive DNA sequences to replicate unevenly, making bodies more sensitive to genetic alterations [75]. MSI is a phenomenon that can manifest in adenomatous and serrated polyps (Figure 2). It is closely associated with alterations in DNA mismatch repair genes resulting from either germline mutations, such as those observed in hereditary nonpolyposis colorectal cancer, or sporadic mutations resulting from abnormal methylation of the MLH1 promoter region. The latter is closely linked with the CpG island methylator phenotype [76]. 

## 4. Dendritic Cells and the Regulation of Immune Response in CRC

The immune response can be initiated and directed by dendritic cells (DCs). Antigens are presented by APCs. In lymphoid as well as non-lymphoid organs, DCs reside or migrate in lymphoid and non-lymphoid subgroups. DCs can seize, digest, and display antigens to T cells. DCs are subtypes that exhibit variations in their differentiation, genetic expression profile, anatomical site, observable characteristics, and purpose [77]. Recently identified DCs include plasmacytoid DCs (pDCs), type 1 conventional DCs (cDC1s), type 2 conventional DCs (cDC2s), Langerhans cells, and monocyte-derived DCs (MoDCs). Under normal circumstances, DCs are chiefly immature, gathering antigens and generating only a few co-stimulatory molecules and effector cytokines. However, when pathogens and injured tissues are present, DCs are activated more quickly. During this stage, their antigen-capturing capacity is reduced, MHC class I and class II antigen expression is increased, and there is a greater secretion of effector cytokines and co-stimulatory molecules, as well as an increased movement to lymph nodes. In these nodes, DCs engage with naive CD4^+^ and CD8^+^ T lymphocytes. By modifying the immune system’s response to malignancies and delivering tumor antigens to T cells, DCs can reduce tumor growth [46]. Numerous studies have revealed that the tumor microenvironment’s complex and contradictory molecular and cellular factors can considerably alter these cells’ phenotypic behaviors. According to extensive research, tumors hinder DCs from developing and boosting their immunosuppressive properties [78]. Alwarawrah et al. found that obesity and diet affect the immune system and cancer progression [79]. There is a relation between the expression of CD11c^+/^CD1c^+^ cells and BMI, even if the amount of CD141^+^/CD11c^+^ DC is the same in slim and overweight people. According to Bertola et al., the accumulation is associated with an increased predominance of Th17 lymphocytes in adipose tissue, signifying that DCs that infiltrate adipose tissue (AT) may regulate tissue inflammation and the growth of Th17 cells [80]. 

Studies utilizing obese mouse models demonstrated that CD131^+^ DC presence in the adipose tissue of lean mice may play an important role in the regional expansion of T regulatory cells, which provide anti-inflammatory signals to preserve adipose tissue homeostasis [81]. Including immune cells in adipocyte-conditioned medium from obese and CRC patients has been observed to increase IL-10 production, decrease DC immunostimulatory function, and inhibit ex vivo T cell-mediated responses. These findings imply an AT microenvironment that regulates and suppresses obesity and CRC [82]. Obesity and CRC alter the peripheral immune cell repertoire [83].

## 5. Alteration in DCs Phenotypic Profile in CRC Patients

Numerous studies have explored the possibility of a relationship between DCs and patients’ clinical responses and have documented both qualitative and quantitative changes in DCs in the blood and tumor microenvironment of CRC patients at various phases of the cancer [78]. Given the appearance of contradicting results in some instances, which might be due to variances in the clinical contexts and procedures used to identify diverse DC subsets, understanding the overall outcomes may be complex. The current study restricted its focus to a sample of selected studies for their exceptional ability to inform clinical practice. Gulubova and his colleagues looked into possible connections between patient prognosis and clinical response and the existence and maturation characteristics of the tumor-infiltrating DCs [84]. According to a research, individuals with advanced illness had a lower frequency of CD83+ mature DCs in the tumor stroma. Another literature source [85], ndicates that there is often a negative relationship between the detection of DCs infiltrating the tumor, metastases to lymph nodes, and the survival of patients with CRC. According to Schwaab et al. [86], many more mature DCs invade primary CRC tissues than the normal colon mucosa. In contrast, the density of DCs in CRC metastases is much lower than in the central tumors. According to Michielsen et al., a tumor-conditioned medium made from cultured human CRC tissue may hinder the development of DCs [87]. This might be explained by the production of chemokines and other soluble substances that could prevent DCs from secreting IL-12p70. In contrast to tumors with microsatellite stability in CRC, Bauer et al. discovered that tumors with high microsatellite instability (MSI-H) have greater levels of mature DC infiltration [88]. This previous finding is interesting because it may shed light on how MSI-H CRC patients respond clinically to novel immunotherapeutic treatments such as CPI (checkpoint inhibitors) [89]. According to Orsini et al. [90], much research has considered the amount and features of DCs in the peripheral blood of CRC patients, compared to normal people, and the cancer phase. According to research, there is a decrease in the amount and effectiveness of monocyte-derived derived-DCs subsets in people with CRC. This shows that the degree to which these effects are felt is closely related to the disease’s stage, the expected result, and the stage of escape (tumor establishment and progression) [91]. Parallel results were discovered by Orsini et al. [90], who noted that patients with complete and advanced-stage CRC had significantly fewer DCs than healthy persons. The research also showed that this decrease was fully recovered once the tumor was entirely removed, supporting the idea that the tumor had a systemic immunosuppressive effect on immune cells in the blood. According to several experts, changes in the pDC population were mainly to blame for the decline in DCs. Granulocyte-macrophage colony-stimulating factor (GM-CSF), various cytokines, such as interleukin-4 (IL-4), interferon (IFN), and other activation and maturation cytokines are used to create monocyte-derived dendritic cells (MoDC), which provide a useful in vitro model for studying the phenotypic profile of DCs and the mediators and mechanisms that are essential in determining their functions. The phenotypic profile of MoDC in patients was compared with control in published research [90]. Orsini et al. showed that people with CRC had a defective differentiation of monocytes into immature DCs in vitro, compared to those without CRC. It is worth mentioning that CRC-MoDC showed diminished costimulatory molecule expression and a reduced ability to deliver antigens to allogeneic T cells and trigger proliferation [59]. Additionally, it exhibited an immunosuppressive cytokine profile primarily described by high levels of IL-10 excretion and decreased IL-12 excretion (Figure 3) [90]. Compared to in vivo blood DCs collected from the same cohort, studies indicated that the maturation status of monocyte-derived DCs (MoDCs) produced from individuals with CRC exhibit superior morphological and functional attributes. This discovery supports the potential efficacy of utilizing MoDCs derived from CRC patients in clinical investigations aimed at cancer immunotherapy [92].

### 5.1. The Role of T Lymphocyte and IL-17 in CRC

Patients who suffer from colorectal cancers that have spread and exhibit repair deficits, along with an elevated density of cytotoxic T cells within the primary tumor, have demonstrated a heightened response rate to immunotherapy based on checkpoint blockade [93]. Immunosuppressive ligands inside the tumor microenvironment may prevent the activation of cytotoxic cells. However, checkpoint blockade treatment has been shown to reduce this inhibition, allowing the cytotoxic lymphocytes to eradicate tumor cells efficiently. Even though cytotoxic T cells have been found within initial colon cancers, some individuals with metastatic CRC resist immunotherapy. The development of the tumor and how it responds to immunotherapeutic treatments may depend on other immune infiltration features, such as the kind of adaptive immune response to the tumor. Extensive research has been conducted on the adaptive immune response to CRC, with a particular emphasis on the functions of various immune cells such as tumor-infiltrating T-regulatory cells (Treg), T-follicular helper (Tfh) cells, Th17 cells, Th1 cells, CD8^+^ and T cells in the progression of CRC. Despite mixed results, a correlation has been established between patient outcomes and the gene expression patterns of Th1 and Th17 cells within malignancies [94]. The expression of IL-17A (both RNA and proteins) was enhanced in around 66% of primary sporadic CRCs. On the other hand, other instances showed elevated interferon-gamma (IFN-γ) expression or a co-expression of IL-17 and IFN-γ [63]. Increased gene expression levels of cytotoxic T cells (PRF1 and GZMB) and Th1 cells (IFN-γ and TBX21) are linked to better patient outcomes, according to studies on the gene-expression profiling of CRC tissues. On the other hand, amplified levels of IL-17A and RORC messenger RNAs, transmitted by Th17 cells, have been related to negative effects on people [64]. There is a great deal of argument about the predictive significance of detecting innate lymphoid T cells, or NK cells, that produce IL-17A in the neoplastic microenvironment, instead of Th17 cells. Little research has shown a significant association between the number of IL17+ cells inside the intra-epithelial compartment and longer periods of relapse-free survival. In the intra-stromal compartment, however, no such connection was seen. However, a separate analysis found that among 125 patients with colorectal cancer, a Th17-cell gene expression pattern was associated with shorter disease-free survival times [95]. The presence of CD8^+^ cytotoxic T cells, CCR5^+^, and CCR6^+^ in malignant tissue is facilitated by epithelial Th17 cells’ secretion of CCL20 and CCL5. This phenomenon is thought to have antitumor properties [94]. However, IL17A also promotes the stromal production of substances that support tumor cell survival, proliferation, and angiogenesis [94]. Patients with colorectal tumors, ranging from adenomas to carcinoma, have higher expression of IL-17A in the stromal and epithelial portions of these lesions [96]. Unfavorable prognoses and increased vulnerability to CRC have been associated with genetic polymorphisms in the genes that encode for the IL-17A, IL-17E, and IL-23 receptors, which are released during the development of Th17 cells. The six members of the IL-17 family (17A, 17B, 17C, 17D, 17E, and IL-17F) can regulate the immune system and send signals through a heterodimeric receptor [97]. According to several studies, IL-17A and IL-17F have the highest proportion of structural homology (55%), and their action method requires binding to the IL-17RA-IL-17RC heterodimer. Various types of cells, including Th17, Tc17 (a subset of CD8^+^ cells), γδT17 cells (IL-17-producing γδT17 cells), and innate immunity lymphoid cells type 3 cells, can produce the cytokines IL-17A and IL-17F. It has been demonstrated that human colorectal carcinomas harbor these cells [97]. Upon binding to the IL-17RA-IL-17RC heterodimer, IL-17A or IL-17F stimulate mitogen-activated protein kinase and NF-kB signaling pathways, triggering the recruitment of adaptor proteins ACT1 and TRAF6. The IL-17RA-IL-17RC complex exhibits variable affinities when binding with homodimers and heterodimers of IL-17A and IL-17F.The binding of IL-17A-IL-17A is shown to have the greatest affinity. It has been shown that myeloid-derived suppressor cell recruitment, T17 cell infiltration, and increased invasiveness are related phenomena [80,97]. Increased expression of vascular endothelial growth factor and micro-vessel density in colorectal cancers have been related to increased levels of IL-17A, indicating a connection between IL-17A and angiogenesis [98]. Human CRC samples show lower levels of IL-17F messenger RNA and IL-17F protein than non-tumor tissues from the same patient. The unique effects of IL-17A and IL-17F signaling via various receptors, IL-17A and IL-17F dimerization, or differences in the expression patterns of IL-17RA and IL-17RC on the carcinogenesis of CRC are poorly understood. The structural integrity of the intestinal barrier is known to be maintained by the binding of IL-17C to IL-17RA-IL-17RE [97]. The receptor complex comprising IL-17RA and IL-17RB can potentially connect with the cytokine IL-17E, which is also recognized as IL-25 and can be produced by both epithelial and myeloid cells. The involvement of IL-17E in stimulating the generation of Th2 cells and activating type 2 innate lymphoid cells has been suggested, particularly in the setting of antiparasitic immune responses at mucosal surfaces [97].

### 5.2. Role of C-Type Lectin Receptor of Pattern Recognition Receptor of Immature DCs in CRC

Myeloid cells include CLRs, a subclass of pattern recognition receptors (PRRs). Damage-associated motif patterns (DAMPs)-detecting receptors activate innate and adaptive immune responses. Pathogen-associated motif patterns (PAMPs) and DAMP-stimulated CLR signaling recruit immune cells and produce cytokines. A recent study has linked many CLRs to intestinal inflammation, a significant risk factor for CRC. CLRs link bacteria, the intestinal epithelial barrier, and the immune system. Fungus identification stimulates the CLR family [98]. PRRs can identify PAMPs and help APCs detect foreign compounds. When exposed to microorganisms, pathogen-associated molecular patterns (PAMPs) activate signaling pathways that enable the immune system to distinguish between different microbes and regulate immune responses accordingly [99]. CLRs are crucial to intestinal fungal immunity, but they also communicate and cooperate with other PRR family members like Toll-like receptors (TLRs), nucleotide-binding oligomerization domain (NOD)-like receptors (NLRs), and RNA-detecting retinoic acid-inducible gene-I (RIG-I)-like receptors (RLRs) [99]. Twenty-seven to forty-one CLR families have cytoplasmic signaling motifs. ITAM-coupled immunoreceptors engage with adaptors with tandem repeats of ITAMs or YxxL in their cytoplastic tails. FcRg and Mincle are the most prevalent CLR-mediated downstream signal transduction ITAM-containing adaptor proteins. The dectin-1 cytoplasmic YxxL motif has one tyrosine (Y) [100]. ITAM or hemITAM tyrosines are phosphorylated when CLRs bind to ligands to attract tyrosine kinase (Syk) in spleen and create complexes such as the CARD9/Bcl-10/MALT1, which activates NF-kB and influences innate and adaptive immune activities. ITAM or hemITAM CLRs can promote and suppress immune responses [101]. Immunoreceptor tyrosine-based inhibitory motif (ITIM)-containing CLRs like DCIR recruit SHP-1 or SHP-2 to impede PRR signaling. LOX-1 and DC-SIGN are human ITAM-ITIM-independent CLRs. CLR intracellular structural patterns are essential for molecular signaling pathways, but they cannot predict immunological reactions or signaling cascades [100]. Context, the location of receptor, multimerization of CLR, the type of ligands and titer, and signaling pathway flexibility are affected by PRR crosstalk [100].

#### 5.2.1. Mincle

Mincle must first interact with the ITAM- FcRg complex to start a regression of signaling cascade that recruits Syk, creates a CARD9/Bcl-10/MALT1 complex, activates NF-kB, and generates cytokines, chemokines, and inflammatory cells [100]. Mincle detects PAMP/DAMP. Adaptable microorganism pathogens and commensals may create PAMP-like a-mannans and glycolipids. Mincle may recognize mycobacterial cell walls’ trehalose-6, 6′-dimycolate (TDM) glycolipid. Murine Mincle modulates intestinal antimicrobial protein synthesis (e.g., RegIIIg, IgA), innate lymphoid cells (ILC) and T cell IL-17 and IL-22 release, and abnormal microbial translocation [102]. The CX3CR1+ gut-resident mononuclear phagocyte (MNP) is a novel Syk-dependent regulator of fungal-host intestinal immune interactions where Mincle, Dectin-1, and Dectin-2 are mainly fungi-recognizing genes. Recent studies show that Mincle signaling causes intestinal inflammation. Inflamed intestinal lamina propria macrophages express Mincle. When wounded tissues release SAP130, active Mincle signaling in macrophages creates inflammatory cytokines via macrophage pyroptosis and draws neutrophils via chemokines such as Cxcl2 and IL-8. Reduced macrophage pyroptosis and proinflammatory cytokines and chemokines enhance experimental colitis. A Mincle inhibitor has not been developed. However, an anti-Mincle neutralizing antibody was shown to improve experimental colitis [103].

#### 5.2.2. Dectin-3

MCL (Macrophage C-type lectin), or Dectin-3, is a transmembrane CLR found on monocytes, neutrophils, macrophages, and DCs. TNF-α upregulates macrophage Dectin-3, Dectin-2, and Mincle. Intestinal fungal dysbiosis significantly increases C. albicans’ load and activates ILC3 to create IL-22 under Stat3, AhR, and IL-7 regulation. IL-22 induces intestinal epithelial cell p-STAT3 carcinogenesis. Mycobiota/Dectin-3/IL-22 regulates colorectal cancer. High fecal fungal load and advanced CRC patients showed decreased Dectin-3 expression, increased IL-22 in malignant tissue, and overall survival. Dectin-3 and Dectin-2 stimulate Kupffer cells to phagocytose cancer cells, preventing cancer cells from liver metastases [104].

#### 5.2.3. DC-SIGN

DC-SIGN detects specific tumor antigens on CRC malignant cells to trigger robust antitumor effects and enable malignancies to elude immunosurveillance. IEC (intestinal epithelial cells) glycosylation changes during malignant transformation. It enhances Lewis Y and Lewis X on CEA specific for tumors, which DC-SIGN recognizes preferentially to facilitate DC-tumor cell contact. Due to low CEA Lewis antigens, normal IECs cannot bind to DC-SIGN. Some primary CRCs produce Mac-2BP, a glycosylated DC-SIGN ligand. CRC-specific glycosylation and DC-SIGN, prevent MoDCs from maturing and differentiating. IL-6 and IL-10 secretions increase to create a CRC-tolerogenic milieu. DC-SIGN interacts with Serine-II Protease MSPL/TMPRSS13 on glycan-free cancer cells. New research shows the DC-SIGN effect on colorectal cancer. Cancer cells infiltrated with DCs upregulate DC-SIGN in tumor tissues, leading to more aggressiveness and invasiveness, poorer prognoses, and a decreased percentage of survival of metastasis-free patients [105]. DC-SIGN activation inhibits miR-185 transcription and enhances VEGF and MMP-9 transcription toward the pathway of PI3K/Akt/b-catenin, promoting CRC metastasis [105]. Metastatic colorectal cancer’s DC-SIGN signaling pathway offers a new etiology and targets for invasion and metastasis. Complex branching N-glycan overexpression helped CRC tumor cells avoid immune detection and create an immunosuppressive environment with decreased IFN-γ production and increased Treg. Intriguingly, eliminating CRC cell-branching N-glycans may reveal immunogenic epitopes of glycan to immune cells via DC-SIGN, boosting anticancer immunity. DC-SIGNR, another DC-SIGN member, may enhance colon cancer hepatic metastasis [106].

#### 5.2.4. LOX-1

Monocytes, macrophages, DCs, B cells, vascular endothelial cells, smooth muscle cells, intestinal cells, cardiomyocytes, adipocytes, platelets, and chondrocytes express LOX-1, a scavenger receptor. Cytokines, oxidized lipoprotein of low-density (ox-LDL), angiotensin II, endothelin, and asymmetric dimethylarginine can upregulate LOX-1 under physiological conditions [107]. LOX-1 binds to Staphylococcus aureus and E. coli. Atherosclerosis is linked to Ox-LDL-LOX-1 interactions. Ox-LDL activates MAPK/NF-kB to increase LOX-1, DC maturation, and pro-inflammatory cytokines. Intriguingly, a recent study indicated that patients with reduced tumor stroma CD8^+^ cytotoxic T lymphocytes (CTL) and LOX-1 had a worse prognosis. Stromal cells that express LOX-1 are mostly M2 macrophages CD163^+^, which determine the cancer prognosis [107]. LOX-1 expression may affect CRC microenvironment M2 macrophages and anticancer immunity. LOX-1 increases tumor development, invasion, and immunosuppression, allowing tumor cells to avoid immune monitoring.

## 6. DCs Utilization in CRC Immunotherapy

The administration of a deceased bacterial vaccination to cancer patients during William Coley’s 1891 trial marked the advent of cancer immunotherapy, which has experienced fluctuations between sanguinity and despondency. Cytokines like IL-2 and IFN-γ have been used clinically to identify human tumor antigens developments in cancer immunotherapy, using adoptive cell treatment procedures. Cancer immunotherapy is now considered innovative. Immune checkpoint inhibitors (CPIs) treatment’s extraordinary results in many cancer patients explain this occurrence. Anti-PD1, Anti-CTLA-4, and anti-PD-L1 antibodies are promising new immunotherapy agents that show relatively minor adverse therapeutic effects in CRC patients [108]. These drugs treat melanoma, non-small lung cancer, and Hodgkin lymphoma well. DCs have been utilized extensively in clinical trials for cancer immunotherapy over a span of two decades, due to their ability to connect innate and adaptive antitumor immunity. GM-CSF (granulocyte macrophage-colony stimulating factor) and IL-4 were used on peripheral blood monocytes to differentiate them in vitro. Before therapeutic vaccination regimens, DCs were given tumor-derived antigens and matured in vitro. The 2011 licensing of the Washington, D.C.-based Provenge prostate cancer vaccine raised hopes for dendritic cell-based cancer vaccines. Cancer vaccine development stalled. DC-based vaccinations were less prioritized than CPI and CAR-T adoptive cell transfer, since hundreds of clinical trials showed a minimal adverse response. DC-based clinical trials have been extensively evaluated. These evaluations also critically assess Field’s main DC clinical development challenges [108,109]. No significant toxicity is a good start, but many important challenges remain. To address concerns related to cellular products, it is essential to establish consistent and dependable criteria for assessing their quality and effectiveness. Furthermore, it is imperative to ascertain the source and quantity of tumor antigens within these products, choose the best injection methods, and explore potential synergies with other pharmaceuticals/interventions to improve medical efficacy. Next-generation DC-based vaccines are being developed again, due to a greater knowledge of DC biology and the discovery of new immunomodulatory drugs that may improve cancer immunotherapies [109]. Recent research suggested testing monocyte-derived DCs with a brief in vitro IFN-γ and GM-CSF exposure for cancer immunotherapy. IFN-DC DCs can internalize apoptotic tumor tissue and induce a strong T lymphocyte subset immune response. Tested phase models and cancer patients showed this. Pilot clinical trials of intratumoral IFN-DC in metastatic melanoma and indolent lymphomas showed promise. Most pilot phase I-II trials examined a small number of metastatic CRC patients. In vitro production of mature DC vaccines through provoking monocytes, using many cytokines for activation and maturation, achieved results worth comparing. Unloaded and laden DCs were used to deliver tumor antigens. Dendritic cell injection protocols and numbers varied widely in several studies. Patients received standard chemotherapy and experimental cell treatments. Dendritic cell-induced immunogenicity evaluation protocols and immuno-monitoring approaches differed. The trials showed that DC-based vaccinations could induce antitumor immune responses in CRC patients without harm. However, utilizing DC to develop innovative CRC immunotherapy techniques requires coordinated research. An immunosuppressive tumor microenvironment, usually found in advanced CRC patients, limits immunotherapy’s efficacy. Successful cancer immunotherapy protocols must address the many complex immunosuppressive pathways in the tumor microenvironment. Macrophage subtypes, T reg cells, and soluble factors are immunosuppressive mediators. Regional cytokine production and responses may contribute to the antitumor immune response type [108,109]. CRC tumor microenvironment immunosuppression is linked to type I IFN receptor depletion [110]. Local cytokine generation and response, particularly type I IFN, may improve CRC immune regulation. In the present era of cancer immunotherapy, understanding the immune response and designing stronger combination therapies, perhaps using dendritic cells, to improve clinical efficacy in subpar responders is a major scientific challenge. Anti-PD1 monoclonal antibodies require intratumoral DCs for anticancer efficacy [110]. A recent study showed the importance of DCs infiltrating the tumor microenvironment in cancer patients’ immunotherapy responses. The study found mature DCs in the tumor microenvironment required for anticancer antibody responses. These DCs produce IL-12 [108,109]. After co-cultivation with peripheral blood cells, IFN-DCs rapidly mature and produce more IL-12. Thus, they may improve anti-PD1 therapy. IFN-DCs stimulate CD4^+^ and CD8^+^T-cell-mediated immune responses against malignant antigen-1 in CRC patients at various phases [109]. Based on our extensive preclinical and clinical evidence of IFN-DC, these DCs are important autologous cellular resources for improving advanced DC agents for CRC clinical trials [111]. DC-based therapy could treat CRC patients with autologous DCs. Figure 4 elicits the role of DCs in immunotherapy. Unloaded antigen-presenting cells (APC), alone or combined with apoptosis-inducing drugs, may be injected intratumorally. In carefully selected combination therapy, antigen-loaded DCs can be used in vitro after injection antibodies specific for anti-PD1 or other CPI to boost antitumor effect [109,112].

## 7. Conclusions

This analysis provided insight into the primary causes, stages, and biological mechanisms during CRC development. It noted that CRC onset is heavily influenced by innate immunity. Additionally, this analysis identified potential biomarkers for CRC tumorigenesis, which could enhance early-stage treatment options. Intestinal DCs regulate T-cell responses and mucosal immunity. As current research indicates, the gut microbiota and its gene products also impact intestinal DC function and may affect microbial-host interactions and immunological homeostasis. All causes of CRC affect GIT microflora, either decreasing beneficial ones or flourishing harmful bacteria. The interactions between the bacteria themselves or their toxins with the immune response (mainly DCs) result in tumor progression and the establishment of several stages of the disease. However, various factors, such as species and strain specificity, patient health and immune status, and the local cytokine microenvironment, must be considered when selecting a microbe for therapeutic nutritional intervention for disease. Although many gut microorganisms are believed to have health-promoting properties that may prevent or treat immunopathology, questions remain about the immunomodulatory effects and the rationale for administering gut commensal bacteria to healthy individuals.

## Figures and Tables

**Figure 1 biomedicines-11-03196-f001:**
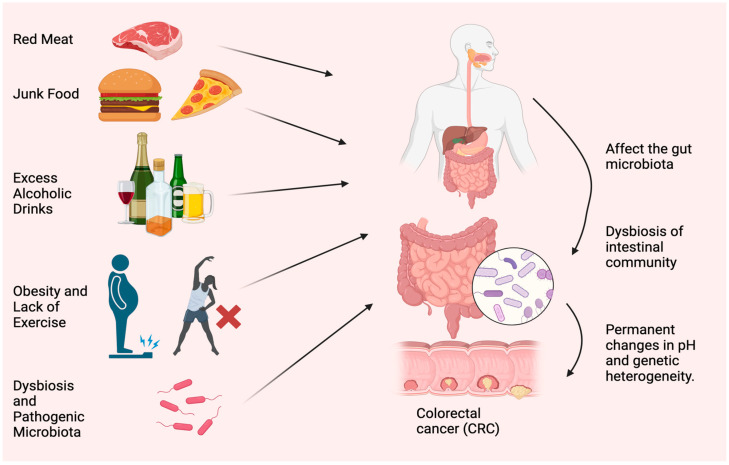
Most typical causes of CRC. Red meat, junk food, excess alcoholic drinks, obesity, lack of exercise, dysbiosis, and accumulation of pathogenic microorganisms are the main reasons for CRC. The figure was designed using BioRender.

**Figure 2 biomedicines-11-03196-f002:**
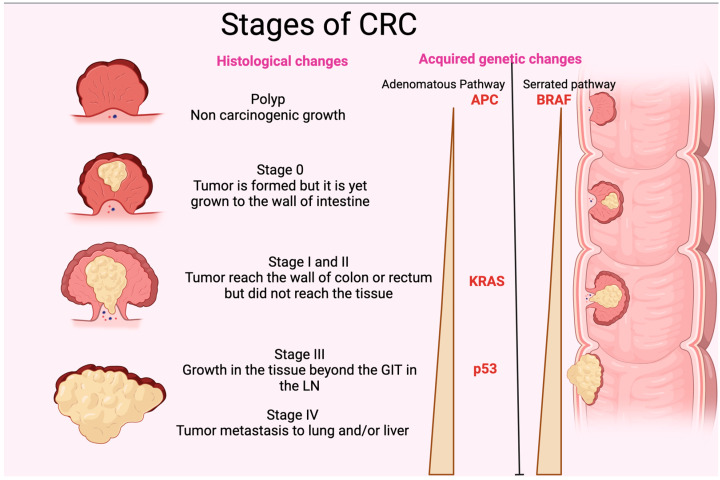
Stages of CRC. The figure shows the different histological stages of CRC. It also represents the gradual increase in p35, KRAS, and APC (adenomatous polyposis coli) in the adenomatous pathway and BRAF in the serrated pathway of acquired genetic changes. GIT: gastrointestinal tract; LN: lymph node. The figure was designed and drawn by BioRender.

**Figure 3 biomedicines-11-03196-f003:**
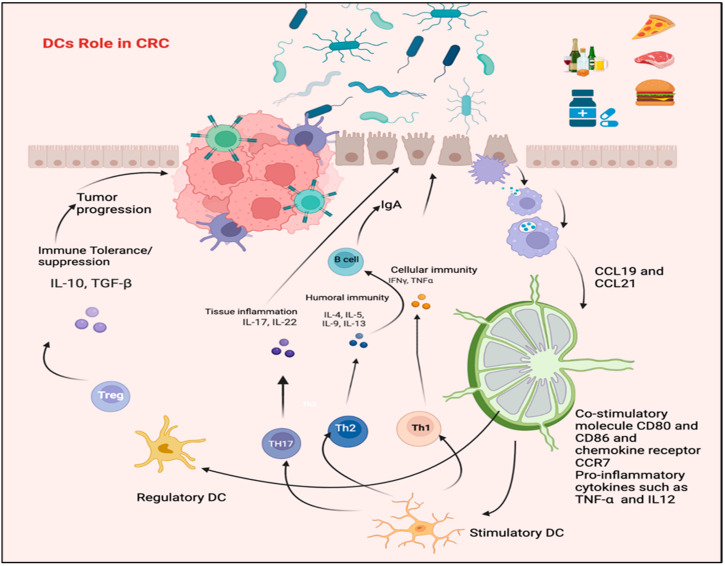
The role of DCs in CRC. Pathogenic bacteria can activate DCs and initiate a protective immune response through Th1/Th17 polarization, possibly leading to IBD. Conversely, DCs can be triggered by microbiota or their gene products to promote stimulation of Tregs, which regulate the functions of Th1 and Th17. The figure was designed and drawn by BioRender.

**Figure 4 biomedicines-11-03196-f004:**
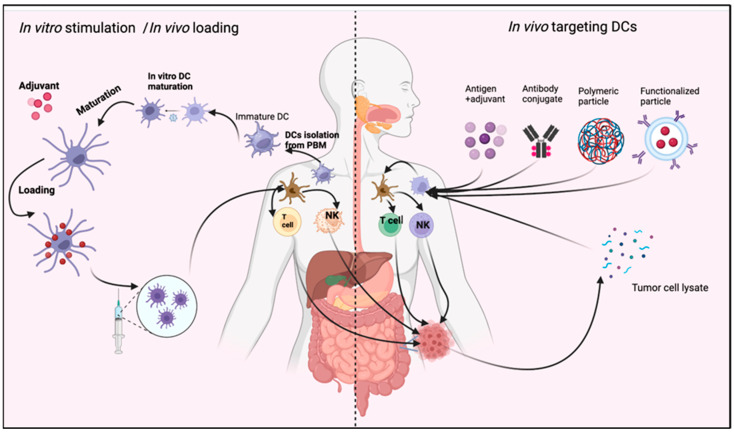
Role of DC in immunotherapy. DCs may be stimulated in vivo through different stimulating agents. Then, mature DCs perform their function as antitumor immune therapy. Another method is in vitro stimulation of isolated DCs from peripheral blood monocyte (PBM). The mature DCs are injected into the blood, with or without a stimulating agent such as adjuvant or tumor cell lysate (TCL). The figure was designed and drawn by BioRender.

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
