# Peer review of "Interaction between Gut Microbiota and Dendritic Cells in Colorectal Cancer"

_biomedicines, 2023, doi:10.3390/biomedicines11123196_

Round 1

Reviewer 1 Report

Comments and Suggestions for Authors

The manuscript suggests that gut microbes influence the delicate equilibrium of dendritic cells during the development of innate and adaptive immune responses to CRC. Although this manuscript compiles some studies on the association between intestinal bacteria and CRC and the possible mechanism of dendritic cells and CRC pathogenesis, the research literature cited in the overall manuscript cannot demonstrate a specific relationship between the two. In addition, the description of intestinal microorganisms and CRC is inadequate, diminishing the article's reference value. Before it is suitable for publication, it must undergo extensive editing.

1. The title should be revised in light of the article's content.

2. The abstract does not correspond to the manuscript's text and must be revised.

3. The conclusion does not provide a sufficient and appropriate explanation of the article's context and should be strengthened.

4. The content of the abstract does not correspond to the content of the conclusion, and the article's central theme cannot be identified.

5. The introduction should emphasize the research purpose of the entire manuscript and explain why it is of academic value to investigate such topics.

6. The cited materials are not sufficiently novel, and the cited materials on some significant research topics are outdated.

7. this research paper aims to highlight the relationship between intestinal bacteria and DC in the pathogenesis of CRC. However, the manuscript's content does not explain the connection between these two topics.

8. Lines 151-153 (ref. 34). This article contains inaccurate information and cannot accurately represent research on pro-carcinogenic bacteria and immune mechanisms.

9. Lines 278 to 285 (ref. 59). The description of the text and cited sources are unclear and muddled. Particularly Line284-285.

10. There are too many keywords, some of which are insufficiently representative.

Author Response

Thank you for your time and effort in reviewing our manuscript and your valuable remarks. We hope that we have done the modification that you suggested. We've written our response to each item after your comment. You will find the modifications in the track changes. Thank you again for your help and support.

1- The title should be revised in light of the article's content.

First, we have tried to clarify that the immune response is initiated by DCs stimulation. First, it is stimulated to produce an inflammatory response by stimulating Th cells. The tumor microenvironment regulates stimulatory DCs and stimulates the production of regulatory DCs to initiate immune tolerance to evade immune response. That’s why DCs are considered the master cell of the immune response. 

Finally, all causes of CRC affect GIT microflora, either decreasing beneficial ones or flourishing harmful bacteria. The interactions between the bacteria themselves or their toxins with the immune response (mainly DCs) result in tumor progression and the establishment of several stages of the disease. 

2- The abstract does not correspond to the manuscript's text and must be revised.

Done. Lines 22-24 and 25-27.

3- The conclusion does not provide a sufficient and appropriate explanation of the article's context and should be strengthened.

Done. Lines 822-825

4- The content of the abstract does not correspond to the content of the conclusion, and the article's central theme cannot be identified.

Done. Lines 25-27.

5- The introduction should emphasize the research purpose of the entire manuscript and explain why it is of academic value to investigate such topics.

Done. Lines 61-65.

5-The cited materials are not sufficiently novel, and the cited materials on some significant research topics are outdated.

CRC was diagnosed as early as 1895. The Topic is not novel, but the correlation between gut microflora and DCs has few citations. Furthermore, we increased the number of references from 80 to 111 (among them 59 references between 2015-2023) and used endnotes to ensure the proper coordination of references and avoid repetition.

6- this research paper aims to highlight the relationship between intestinal bacteria and DC in the pathogenesis of CRC. However, the manuscript's content does not explain the connection between these two topics.

We tried to clarify this point in lines 220-269.

7-Lines 151-153 (ref. 34). This article contains inaccurate information and cannot accurately represent research on pro-carcinogenic bacteria and immune mechanisms.

We removed the reference, corrected the mechanism, and add two references to support this mechanism.

9-Lines 278 to 285 (ref. 59). The description of the text and cited sources are unclear and muddled. Particularly Line284-285.

 Clarification was done. Line 376

10- There are too many keywords, some of which are insufficiently representative.

Done. We deleted some.

Reviewer 2 Report

Comments and Suggestions for Authors

In this review by Zaher and Basingab, the authors highlight the potential role of dendritic cells (DCs) and microbiome in colorectal cancer (CRC) in immunotherapy, with emphasis on innovative approaches meant to highlight the use of DCs (e.g. tumor vaccination). The review is timely, as a recent search of Pubmed using terms "colorectal cancer, dendritic cells, microbiome" yielded only a few reviews. This is an interesting topic of current research. The overall writing is moderate to well, with some instances of improper grammar highlighted in the English quality portion of the review. There are some nice figures, appropriate subsections, but there are some issues that might need to be addressed, particularly in the focus of the review:

1. The title gives the impression that this review will highlight the interactions between the gut microbiome and DCs during colon cancer or CRC. However, the way it is currently written, there is little connection, but rather separate sections. While the authors have a section of the gut microbiome (Section 2.4), how this eventually connects to DCs and their potential role in CRC immunotherapy is not well addressed in the review. What is the purpose of the gut microbiome section? The lack of depth to the gut microbiome in the current report is evident in that it is sparse compared to the other key topic of DCs.

2. In line 56, the authors mentioned "unchangeable risk factors". Type II diabetes can be prevented through healthy lifestyle changes, so how is this unchangeable?

3. Subsection 2.1: What is "junk food" and why is this mentioned in the title? This section focuses on red meat and there are no mentions of studies on "junk food" and CRC.

4. Section 2.4.1: in Line 144, by "germs", do you mean Fusobacterium nucleatum? If so, this should be made more clear.

5. Section 2.4.2: This subsection is just two sentences and not well defined. Perhaps more can be added regarding this topic; or rather, once again, it is not known exactly why the gut microbiome is relevant to this review, as it seems to more focus on DCs.

6. Figure 1: Please define what is "PH" in this context, as it is not highlighted in the manuscript. Also, not certain why "Genetic" is capitalized.

7. Figure 2: Define what GIT and LN are. Also, it is unclear if the "gradual increase" is just for APC in the Adenomatous Pathway, or if this is also for p53 and KRAS.

8. Line 213: The use of APCs is confusing in the review. This could for antigen presenting cells (APCs) or adenomatous polposis coli (APC), a tumor suppressor gene. Please make this clear in the writing.

9. Line 259: Did you mean "nodes" here instead of "neds"?

10. Line 269: Please defined what is meant by CPI.

11. Line 290: it is unclear why the authors started a new paragraph here. It is related to the previous paragraph, so why separate it?

12. Line346, 354: What are "T17 cells"? These are different from Th17 or Tc17 cells? If so, please add more information on what these cells are.

13. Lines 396-397: This is awkward/incomplete sentence. Perhaps it was meant to read as "are affected"?

14. Line 409: Is this meant to be "mincle signaling" instead of "mince"?

15. Lines 414-415: Awkward sentence without clear meaning. Did the authors mean to say that anti-Mincle neutralizing antibody "was shown" to improve experimental colitis?

16. Lines 423-425: This is an incomplete sentence, and what is the purpose of the "(". Not sure what was trying to be conveyed here.

17. Line 471: What the authors say "minor therapeutic effects", did you mean negative or adverse effects? It is unclear from source if this sentence suggests that these therapies are not effective, but the authors state they are promising agents. Line 480 has the same issue with "minimal response" It is unclear if the authors suggest these approaches are effective or lacking adverse side effects associated with them.

18. Lines 538-539: The first sentence is incomplete, and unsure what the authors are trying to illustrate here.

19. The authors declare no funding supported this work (Line 558), but then say it was funded in the Acknowledgements. 

Comments on the Quality of English Language

The quality of English is moderate to well. There are several instances of awkward sentences and departure from standard English language practices (use of abbreviations). Such instances are detailed below:

1. There were several times the authors did not abbreviate colorectal cancer (CRC), despite establishing this in Line 32. Please correct the instances which appear in the following Lines: 45, 57, 78, 83, 93, 99, 100, 160, 173, 180, 265, 304, 314-315, 324, 332, 356, 361, 421-422, 436, 441, 524.

2. Inflammatory Bowel Disease (IBD) was first mentioned in Line 56, and should be abbreviated at the following Lines: 84 (I think this was supposed to be Inflammatory, not Infectious), 155, 300..

3. Dendritic Cells (DCs) was first abbreviated on 219. Please abbreviate where appropriate in the following Lines: 230, 476, 478, 492, 498-500, 513, 517, 519, 523, 541.

4. Line 436: Should not capitalize "Infiltrate"

5. Line 235: By "AT", do the authors mean "adipose tissue"? If so, this should be abbreviated at first mention and used throughout appropriately.

Author Response

Thank you for your time and effort in reviewing our manuscript and your valuable remarks. We hope that we have done the modification that you suggested. Our response to each item is written after you've commented. You will find the modifications in the track changes. Thank you again for your help and support.

1- The title gives the impression that this review will highlight the interactions between the gut microbiome and DCs during colon cancer or CRC. However, the way it is currently written, there is little connection, but rather separate sections. While the authors have a section of the gut microbiome (Section 2.4), how this eventually connects to DCs and their potential role in CRC immunotherapy is not well addressed in the review. What is the purpose of the gut microbiome section? The lack of depth to the gut microbiome in the current report is evident in that it is sparse compared to the other key topic of DCs.

We have tried to clarify that the immune response is initiated by DCs stimulation. First, it is stimulated to produce an inflammatory response by stimulating Th cells. The tumor microenvironment down-regulates stimulatory DCs while stimulating the production of regulatory DCs to initiate immune tolerance to evade immune response. That’s why DCs are considered the master cell of the immune response.  

2- In line 56, the authors mentioned "unchangeable risk factors". Type II diabetes can be prevented through healthy lifestyle changes, so how is this unchangeable?

In fact, it is reversible as long as the immune response is at preliminary stages, but once a lump is formed, it is irreversible and can not be changed.

3- Subsection 2.1: What is "junk food" and why is this mentioned in the title? This section focuses on red meat and there are no mentions of studies on "junk food" and CRC.

This part was explained in detail in lines 92-107

4- Section 2.4.1: in Line 144, by "germs", do you mean Fusobacterium nucleatum? If so, this should be made more clear.

Yes, we mean Fusobacterium nucleatum and we clarified this in line 166

4- Section 2.4.2: This subsection is just two sentences and not well defined. Perhaps more can be added regarding this topic; or rather, once again, it is not known exactly why the gut microbiome is relevant to this review, as it seems to more focus on DCs.

The review is focused on the correlation of gut microbiota and DCs and Colorectal Cancer. We should state certain Germs related to CRC. The correlation between DCs and the bacterial pathogen is clarified in lines 166- 209

6- Figure 1: Please define what is "PH" in this context, as it is not highlighted in the manuscript. Also, not certain why "Genetic" is capitalized.

We mean pH, and the word ‘Genetic’ is corrected to ‘genetic.’

7- Figure 2: Define what GIT and LN are. Also, it is unclear if the "gradual increase" is just for APC in the Adenomatous Pathway, or if this is also for p53 and KRAS.

GIT: gastro-intestinal tract (we defined it in line 299)

LN: lymph node (we described it in line 300). We also clarified the gradual increase of APC at the first stages and the gradual increase of KRAS at large-size tumors, while p53 at advanced stages of the Adenomatous Pathway.

8- Line 213: The use of APCs is confusing in the review. This could for antigen presenting cells (APCs) or adenomatous polposis coli (APC), a tumor suppressor gene. Please make this clear in the writing.

Done Lines: 270 and 298.

9- Line 259: Did you mean "nodes" here instead of "neds"?

Yes, we changed it.

10- Line 269: Please defined what is meant by CPI.

CPI: Checkpoint inhibitor. We defined it.

11- Line 290: it is unclear why the authors started a new paragraph here. It is related to the previous paragraph, so why separate it?

Done.

12- Line346, 354: What are "T17 cells"? These are different from Th17 or Tc17 cells? If so, please add more information on what these cells are.

The definition of each cell was added.

13- Lines 396-397: This is awkward/incomplete sentence. Perhaps it was meant to read as "are affected"?

Done

14- Line 409: Is this meant to be "mincle signaling" instead of "mince"?

Done

15- Lines 414-415: Awkward sentence without clear meaning. Did the authors mean to say that anti-Mincle neutralizing antibody "was shown" to improve experimental colitis?

Yes, and it is done.

16- Lines 423-425: This is an incomplete sentence, and what is the purpose of the "(". Not sure what was trying to be conveyed here.

Done. Lines 523-525

17- Line 471: What the authors say "minor therapeutic effects", did you mean negative or adverse effects? It is unclear from source if this sentence suggests that these therapies are not effective, but the authors state they are promising agents. Line 480 has the same issue with "minimal response" It is unclear if the authors suggest these approaches are effective or lacking adverse side effects associated with them.

Yes, we mean minimal adverse response. It is clarified in lines 570 and 580

18- Lines 538-539: The first sentence is incomplete, and unsure what the authors are trying to illustrate here.

It is corrected in lines 637-638.

19- The authors declare no funding supported this work (Line 558), but then say it was funded in the Acknowledgements. 

Done

Comments on the Quality of English Language

The quality of English is moderate to well. There are several instances of awkward sentences and departure from standard English language practices (use of abbreviations). Such instances are detailed below:

  1. There were several times the authors did not abbreviate colorectal cancer (CRC), despite establishing this in Line 32. Please correct the instances which appear in the following Lines: 45, 57, 78, 83, 93, 99, 100, 160, 173, 180, 265, 304, 314-315, 324, 332, 356, 361, 421-422, 436, 441, 524.

Done

  1. Inflammatory Bowel Disease (IBD) was first mentioned in Line 56, and should be abbreviated at the following Lines: 84 (I think this was supposed to be Inflammatory, not Infectious), 155, 300.

Done

  1. Dendritic Cells (DCs) was first abbreviated on 219. Please abbreviate where appropriate in the following Lines: 230, 476, 478, 492, 498-500, 513, 517, 519, 523, 541.

Done

  1. Line 436: Should not capitalize "Infiltrate"

Done

  1. Line 235: By "AT", do the authors mean "adipose tissue"? If so, this should be abbreviated at first mention and used throughout appropriately.

yes, and it is Done

Round 2

Reviewer 1 Report

Comments and Suggestions for Authors

All concerns have been answered and corrected with appropriate replies.

Author Response

We thank you for your time and effort in reviewing our manuscript. Thanks to your helpful and valuable comments, the review is much better. We are very grateful for your help and support.    

Reviewer 2 Report

Comments and Suggestions for Authors

The authors have addressed the major concern in connecting the gut microbiome component of the review to DCs, which was not as apparent in the original version. Most grammatical errors in the original version have also been corrected.

Comments on the Quality of English Language

Minor concerns in grammar:

1. Line 29: "Checkpoints" should not be capitalized.

2. Lines 25-27. Incomplete sentence. Suggest beginning with "In addition, this review focuses also on the role of microbiota in dysbiosis..."

3. Line 55: Suggest changing "spot the light on the main causes" to "spotlight the main causes".

4. Lines 55-58 (awkward english/incomplete sentences): Suggest the following change, " The purpose of this article is to spotlight the main causes of CRC and how they affect gut microflora. In addition, this review will also focus on how DCs play an important role in tumor regulation and eventually explain the promising role of immunotherapy in CRC treatment control. "

Author Response

We thank you for your time and effort in reviewing our manuscript. Thanks to your helpful and valuable comments, the review is much better. We are very grateful for your help and support.  

  We hope that we have done the modification that you suggested. Our response to each item is written after your comment. You will find the modifications in the track changes. Thank you again for your help and support.

  1. Line 29: "Checkpoints" should not be capitalized.

      Done

  1. Lines 25-27. Incomplete sentence. Suggest beginning with "In addition, this review focuses also on the role of microbiota in dysbiosis..."

Done

  1. Line 55: Suggest changing "spot the light on the main causes" to "spotlight the main causes".

Done

  1. Lines 55-58 (awkward english/incomplete sentences): Suggest the following change, " The purpose of this article is to spotlight the main causes of CRC and how they affect gut microflora. In addition, this review will also focus on how DCs play an important role in tumor regulation and eventually explain the promising role of immunotherapy in CRC treatment control. "

Done